# Early Universe Thermodynamics and Evolution in Nonviscous and Viscous Strong and Electroweak Epochs: Possible Analytical Solutions

**DOI:** 10.3390/e23030295

**Published:** 2021-02-28

**Authors:** Abdel Nasser Tawfik, Carsten Greiner

**Affiliations:** 1Egyptian Center for Theoretical Physics, Juhayna Square off 26th-July-Corridor, Giza 12588, Egypt; 2Institute for Theoretical Physics (ITP), Goethe University Frankfurt, Max-von-Laue-Str. 1, D-60438 Frankfurt am Main, Germany; Carsten.Greiner@th.physik.uni-frankfurt.de

**Keywords:** viscous cosmology, particle-theory and field-theory models of the early universe, mathematical and relativistic aspects of cosmology, thermodynamic functions and equations of state, 98.80.-k, 98.80.Cq, 98.80.Jk, 05.70.Ce

## Abstract

**Simple Summary:**

In the early Universe, both QCD and EW eras play an essential role in laying seeds for nucleosynthesis and even dictating the cosmological large-scale structure. Taking advantage of recent developments in ultrarelativistic nuclear experiments and nonperturbative and perturbative lattice simulations, various thermodynamic quantities including pressure, energy density, bulk viscosity, relaxation time, and temperature have been calculated up to the TeV-scale, in which the possible influence of finite bulk viscosity is characterized for the first time and the analytical dependence of Hubble parameter on the scale factor is also introduced.

**Abstract:**

Based on recent perturbative and non-perturbative lattice calculations with almost quark flavors and the thermal contributions from photons, neutrinos, leptons, electroweak particles, and scalar Higgs bosons, various thermodynamic quantities, at vanishing net-baryon densities, such as pressure, energy density, bulk viscosity, relaxation time, and temperature have been calculated up to the TeV-scale, i.e., covering hadron, QGP, and electroweak (EW) phases in the early Universe. This remarkable progress motivated the present study to determine the possible influence of the bulk viscosity in the early Universe and to understand how this would vary from epoch to epoch. We have taken into consideration first- (Eckart) and second-order (Israel–Stewart) theories for the relativistic cosmic fluid and integrated viscous equations of state in Friedmann equations. Nonlinear nonhomogeneous differential equations are obtained as analytical solutions. For Israel–Stewart, the differential equations are very sophisticated to be solved. They are outlined here as road-maps for future studies. For Eckart theory, the only possible solution is the functionality, H(a(t)), where H(t) is the Hubble parameter and a(t) is the scale factor, but none of them so far could to be directly expressed in terms of either proper or cosmic time *t*. For Eckart-type viscous background, especially at finite cosmological constant, non-singular H(t) and a(t) are obtained, where H(t) diverges for QCD/EW and asymptotic EoS. For non-viscous background, the dependence of H(a(t)) is monotonic. The same conclusion can be drawn for an ideal EoS. We also conclude that the rate of decreasing H(a(t)) with increasing a(t) varies from epoch to epoch, at vanishing and finite cosmological constant. These results obviously help in improving our understanding of the nucleosynthesis and the cosmological large-scale structure.

## 1. Introduction

The current thorough knowledge on the cosmic evolution is primarily based on the standard model of cosmology (SMC), which introduces a generic hypothesis that the cosmic background is isotropically and homogeneously filled up with an exclusively ideal fluid. After all, we simply realize that this is an abstraction, i.e., a general description that is not based on real physical situation. Apart from approaches, models, and theories, the real situation could have arose out from recent high-energy experiments [1,2,3,4,5,6,7] and cosmic observations [8,9,10,11,12,13]. Over the years, it was assumed that the impacts of the viscosity coefficients on cosmology should be weak or at least subdominant so that the inclusion of viscous concepts in the macroscopic theory of the cosmic fluid appeared as most natural improvement. It was first assumed that the influence of viscosity in the early Universe would be the largest at the the end of the lepton era, i.e., during the neutrino decoupling era, at temperature ≃$10^{10}\;$K. Viscous coefficients connected with particle physics have been also proposed by Misner [14,15]. Recent studies reveal that the impact of viscosity likely sets on during the very early stages of the Universe [16]. The present study suggests extending SMC to *beyond* SMC. In bCMS, the cosmic background geometry is filled with viscous matter, whatever its constituents are, so that isotropicity and homogeneity are generalized.

For the inclusion of the viscous properties, one would like to start with small perturbations from the thermal equilibrium. The suitable theoretical framework for this is the first-, Section 4.2, and the second-order cosmic relativitic fluid, Section 4.3. Both viscosity coefficients, the bulk viscosity ζ and the shear viscosity η can be determined. From SMC considerations that the Universe spatially homogeneously expands, and cosmological observations [8,9,10,11,12,13], ζ would be taken as a dominant component, while η would be neglected. In bCMS, this assumption could be also generalized. The motivations for viscous theories in cosmology are diversified. For instance, over the last three decades, various attempts have been reported in literature [16,17,18,19,20]. A direct implementation of the equations of state (EoS) deduced from recent lattice quantum chromodynamic calculations and/or heavy-ion collisions on physics of the early Universe was initiated in various studies conducted by one of the authors [16,20,21,22,23,24,25,26]. The present paper resumes these studies, especially in light of the recent progress which enabled us to explore the very early epochs of the evolution of the Universe [16,27,28,29,30]. The procedure goes as follows. The viscous EoS introduced in Section 4.1 and taken from Ref. [16] shall be substituted in the Friedmann equations. This leads to sophisticated differential equations. Their analytical solutions turn into a very challenging mathematical task. By finding unambitious analytical solutions, bSMC becomes a feasible approach. In this paper, we introduce and discuss the possible analytical solutions; the ones expressing the Hubble parameter in dependence on the scale factor, i.e., functionality H(a(t)), where both quantities are also functions of the cosmic time *t*. We also introduce a road-map for future studies based on bSMC.

The present script is organized as follows. The cosmic geometry and the field equations will be reviewed in Section 2. The cosmic evolution in non-viscous background geometry classified into different epochs will be discussed in Section 3. The cosmic evolution in viscous background geometry, Section 4, is based on viscous EoS introduced in Section 4.1, where the background fluid is described by first-order Eckart theory, Section 4.2 and second-order Israel–Stewart theory, Section 4.3. The results on the possible analytical solutions, i.e., functionality H(a(t)), where both *H* and *a* are also functions of the cosmic time *t*, will be elaborated in Section 5. Section 6 is devoted to draw the final conclusions. Other details can be seen at Appendix A.

## 2. Geometry and Field Equations

In curved cosmic geometry under the assumptions of SMC (homogeneity and isotropy) for cosmic space and matter, the Friedmann-Lemaitre-Robertson-Walker (FLRW) metric reads
(1)$$ds^{2}=dt^{2}-a{\left(t\right)}^{2}\left[\frac{dr^{2}}{1-kr^{2}}+r^{2}\left(d\theta^{2}+\sin^{2}\theta d\phi^{2}\right)\right],$$
where a(t) is the dimensionless scale factor, which describes the expansion of the Universe. *k* characterizes elliptical, flat (Euclidean), and hyperbolic cosmic space, where k={−1,0,+1} stands for negative, flat, and positive curvature, respectively. It should be noticed that if *r* is taken dimensionless, a(t) shall be given in a unit of length. In Equation (Equation 1) and to simplify the cosmology notation, we use the natural units c=G=1. So far, the theory of general relativity does not enter the play. It certainly does, when the evolution of s(t), the temporal evolution of the line element, should be tackled. Towards this end, the theory of general relativity should be combined with the matter/energy content of the space-time within the cosmic geometry.

The Einstein gravitational fields with finite cosmological constant are given as
(2)$$R_{\mu \nu }-\frac{1}{2}g_{\mu \nu }\;R+\Lambda_{\mu \nu }=\frac{8\pi }{3}\;T_{\mu \nu },$$
where the indices μ, ν take discrete values 0, 1, 2, 3. The energy–momentum tensor of the bulk viscous cosmological fluid filling the very early Universe can be expressed as [18]
(3)$$T_{\mu \nu }=\left(\rho +p+\Pi \right)\;u_{\mu }u^{\nu }-\left(p+\Pi \right)g_{\mu \nu },$$
where ρ is the energy density, *p* is the thermodynamic pressure, Π is the bulk viscous pressure, and $u_{\mu }$ is the four velocity satisfying the normalization condition $u_{\mu }u^{\mu }=1$. The bulk pressure Π can formally be included in the thermodynamic pressure $p_{\mathrm{eff}}=p+\Pi$. We shall discuss on how to evaluate Π, concretely the bulk viscous pressure, in framework of Eckart (first-order), Section 4.2, and Israel–Stewart (second-order) theories, Section 4.3, for relativistic viscous cosmic fluid.

For number density *n*, specific entropy *s*, finite temperature *T*, bulk viscosity coefficient ζ, and relaxation time τ, the particle and entropy fluxes are to be related to each other as $N^{i}=\nu^{i}$ and $S^{i}=sN^{i}-\left(\tau \Pi^{2}/2\zeta T\right)u^{i}$, respectively. It should be emphasized that the evolution of the cosmological fluid is subject to the dynamical laws of particle number conservation $N_{\;;i}^{i}=0$ and Gibbs’ equation $Td\rho =d\left(\rho /n\right)+pd\left(1/n\right)$ [18]. In what follows, we assume that the energy–momentum tensor of the cosmological fluid is locally conserved, i.e., $T_{i;k}^{k}=0$, where ; denotes the covariant derivative with respect to the line metric.

In the proper frame, i.e., the inertial frame of reference comoving with the the fluid, the components $T_{0}^{0}=\rho$, $T_{1}^{1}=T_{2}^{2}=T_{3}^{3}=-p_{\mathrm{eff}}$. For the isotropic and homogeneous metric given in Equation (Equation 1), the Einstein field equations in natural units read
(4)$$\begin{array}{ccc} H{\left(t\right)}^{2} & = & \frac{8\pi }{3}\;\rho \left(t\right)-\frac{k}{a{\left(t\right)}^{2}}+\frac{\Lambda }{3}, \end{array}$$
(5)$$\begin{array}{ccc} \dot{H}\left(t\right)+H{\left(t\right)}^{2} & = & -\frac{4\pi }{3}\;\left[\rho \left(t\right)+3p_{\mathrm{eff}(t)}\right]+\frac{\Lambda }{3}, \end{array}$$
where the dot refers to differentiation with respect to the cosmic time *t*, and $H\left(t\right)=\dot{a}\left(t\right)/a\left(t\right)$ is the Hubble parameter. From the expressions (Equation 4) and (Equation 5), the time evolution of Hubble parameter can be deduced as
(6)$$\begin{array}{ccc} \dot{H}\left(t\right) & = & -4\pi \;\left[\rho \left(t\right)+p_{\mathrm{eff}}\left(t\right)\right]+\frac{k}{a{\left(t\right)}^{2}}. \end{array}$$

From the local conservation of the energy–momentum tensor in the Universe, the following equation has been proposed by McCrea and Milde and by Peebles with vanishing [31,32] and finite pressure [33], respectively, as being equivalent to Newtonian mechanics,
(7)$$\dot{\rho }\left(t\right)+3\left[\rho \left(t\right)+p_{\mathrm{eff}}\left(t\right)\right]H\left(t\right)=0.$$

This means that the decrease in the energy content of a cube with side a(t) equals the energy budget due to the expansion of the Universe and the work done by the pressure on the surface.

To obtain a closed system of equations, we need to propose EoS relating *p* to ρ. Depending on the approach we are applying, we might also need to propose a reliable estimation for Π, as well. In the section that follows, we introduce solution for the Friedmann equation based on various types of EoS. We first assume vanishing bulk viscosity, Section 3. Then, we discuss on the extension to finite bulk viscosity, Section 4, which then required barotropic equations for the pressure, Equations (Equation 8)–(Equation 10), the temperature, Equation (Equation 37), the bulk viscosity coefficient, Equations (Equation 34)–(Equation 36), and the relaxation time, Equations (Equation 39)–(41). These are examples on novel contributions presented by the present script.

## 3. Cosmic Evolution in Non-Viscous Approach

By combining recent non-perturbative and perturbative calculations with other degrees of freedom (dofs), such as photons, neutrinos, leptons, electroweak particles, and Higgs bosons, various thermodynamic quantities for almost net-baryon-free cosmic matter have been calculated up to the TeV-scale, i.e., covering quantum chromodynamic (QCD) and electroweak (EW) eras of the early Universe [16]. It was found that while the EoS relating the pressure *p* to the energy density ρ for the hadronic matter is simple, the one for QCD and that EW matter are rather complicated. It is worth highlighting that in the cosmological context, the various thermodynamic quantities should be translated into time-depending quantities. When the cosmic time elapses, the spacial dimensions of the Universe expand, and accordingly thermodynamic quantities characterizing the background geometry vary. The EoS proposed in Figure 1 [16] are preliminary depending on the energy density, whose decrease might be—for simplicity—taken as a scale for increasing cosmic time and vice versa.
(8)Hadron:p(t)=α1+β1ρ(t),(9)QCD/EW:p(t)=α2+β2ρ(t)+γ2ρ(t)δ2,
where $\alpha_{1}=0.0034\pm 0.0023$, $\beta_{1}=0.1991\pm 0.0022$, $\alpha_{2}=0.0484\pm 0.0164$, $\beta_{2}=0.3162\pm 0.031$, $\gamma_{2}=-0.21\pm 0.014$, and $\delta_{2}=-0.576\pm 0.034$. At very large ρ(t), the asymptotic behavior becomes very close to that of an ideal gas limit,
(10)$$\begin{array}{ccc} \mathrm{Asymp}.:\;\;p(t) & = & \gamma_{3}\rho \left(t\right), \end{array}$$
where $\gamma_{3}=0.3304\pm 0.0236$. It should be noticed that the EoS (Equation 8), in the hadronic phase, with its positive parameters $\alpha_{1}$ and $\beta_{1}$ could easily be—due to its large uncertainty—reexpressed with vanishing $\alpha_{1}$. Nevertheless, in the present calculations, we keep $\alpha_{1}$ finite. In Section 3, we present solutions for the Friedmann equations, Equations (Equation 4)–(Equation 6), with the various Equations (9) and (Equation 10), which, as mentioned, characterize various types of cosmic backgrounds corresponding to various epochs of the early Universe. The present paper introduces numerical results obtained for the analytical dependence of the Hubble parameter on the scale factor.

At vanishing bulk viscosity, Π(t)=0, the effective pressure $p_{\mathrm{eff}}\left(t\right)=p\left(t\right)+\Pi \left(t\right)$ can be simplified as the thermodynamic pressure p(t). Then, the Friedmann Equation (Equation 6) can be rewritten as
(11)$$\begin{array}{ccc} \ddot{a}\left(t\right)\;a\left(t\right)-\dot{a}{\left(t\right)}^{2}+4\pi \left[\rho (t)+p(t)\right]a{\left(t\right)}^{2}-k & = & 0, \end{array}$$
which can be solved if combined with set of closed equations, such as Equation (Equation 4) and suitable EoS. Accordingly, we have various solutions characterizing the various eras in the early Universe.

### 3.1. Hadronic Era

By substituting Equations (Equation 4) and (Equation 8) into Equation (Equation 11), we get
(12)$$\begin{array}{ccc} \ddot{a}\left(t\right)\;a\left(t\right)+C_{1}\;\dot{a}{\left(t\right)}^{2}+C_{2}\;a{\left(t\right)}^{2}+C_{1}\;k & = & 0, \end{array}$$
where the variables $C_{1}$ and $C_{2}$ are given in Table 1. $C_{1}$ and $C_{2}$ are functions of the coefficients obtained in the parameterized EoS, which in tern vary from epoch to epoch. For the sake of simplicity, the coefficients are conjectured remaining constant within each epoch. We notice that the value of $C_{1}$—in the hadron era—is finite but not necessarily unified. This assures that *k*, the curvature contact, remains finite.

At vanishing *k*, which is the case at $\beta_{1}=-1/3$, we have analytical solutions. Then, Equation (Equation 12) can be solved as,
(13)$$\begin{array}{ccc} a(t) & = & c_{2}\cosh{\left[\sqrt{C_{2}\left(1-C_{1}\right)}\left(t+c_{1}\right)\right]}^{1/(1-C_{1})}, \end{array}$$
where $c_{1}$ and $c_{2}$ are integration constants which can be fixed at boundary and initial conditions. For instance, at t=0, H(t)=0, Equation (Equation 14), then, $c_{1}=-t$. In general, $c_{1}$ has the dimension of the cosmic time *t* and therefore varies with the evolution of the Universe. Hence, $c_{1}$ is finite and the cosmological parameters of the hadron epoch (assuming that $6.58\times 10^{-16}\;s={\mathrm{eV}}^{-1}$) $∼15.197\times 10^{14}\;$MeV${}^{-1}$), for instance Equation (Equation 14) can be estimated, numerically. On the other hand, for the scale factor, a(t), we still need to estimate the other integration constant $c_{2}$. Having an analytical expression for the scale factor, Equation (Equation 13), then the Hubble parameter can then be obtained,
(14)$$\begin{array}{ccc} H(t) & = & \frac{\dot{a}\left(t\right)}{a(t)}=\sqrt{\frac{C_{2}}{1+C_{1}}}\;\tanh\left[\sqrt{C_{2}\left(1+C_{1}\right)}\left(t+c_{1}\right)\right]. \end{array}$$At non-vanishing *k*, there is no direct analytical solution for a(t). When assuming that $u={\dot{a}}^{2}\left(t\right)$ and substituting this into Equation (Equation 12),
(15)$$\begin{array}{ccc} \frac{du(a(t))}{da(t)}+2C_{1}\;\frac{u(a(t))}{a(t)}+2C_{2}a\left(t\right)+2C_{1}\frac{k}{a(t)} & = & 0, \end{array}$$
a solution for $\dot{a}\left(t\right)$ can be proposed
(16)$$\begin{array}{ccc} u(a(t)) & = & \dot{a}{\left(t\right)}^{2}\\ & = & c_{1}a{\left(t\right)}^{-2C_{1}}-\frac{C_{2}}{1+C_{1}}a{\left(t\right)}^{2}-\frac{k}{C_{1}+1}a{\left(t\right)}^{2}. \end{array}$$The physical solution is the one assuring that,
(17)$$\begin{array}{ccc} a(t) & < & {\left[\frac{c_{3}}{C_{2}}\left(1+C_{1}\right)\right]}^{\frac{1}{2(1-C_{1})}}. \end{array}$$Hence, the Hubble parameter can be deduced as
(18)$$\begin{array}{ccc} H(t) & = & {\left\{c_{1}a{\left(t\right)}^{-2C_{1}}-\frac{C_{2}+k}{C_{1}+1}a{\left(t\right)}^{2}\right\}}^{1/2}\frac{1}{a(t)}. \end{array}$$By solving the second-order differential Equation (Equation 16), an analytical expression for the scale factor a(t) can also be deduced,
(19)$$\begin{array}{ccc} a(t) & = & {\left[\sqrt{\frac{C_{2}+k}{c_{1}\left(C_{1}+1\right)}}\right]}^{\frac{1}{-(1+C_{1})}}, \end{array}$$
whose time dependence is given by the time dependence of the corresponding EoS, namely $C_{1}$ and $C_{2}$, which are listed in Table 1. As discussed, they have an indirect time dependence through the coefficients of the corresponding EoS. Within one era, they are conjectured to remain constant. The latter might be the only way possible to gain an analytical solution. Then, H(t), Equation (Equation 18), can be rewritten as
(20)$$\begin{array}{ccc} H(t) & = & c_{1}{\left[{\left[\sqrt{\frac{C_{2}+k}{c_{1}\left(C_{1}+1\right)}}\right]}^{-\frac{1}{1+C_{1}}}\right]}^{-(1+2C_{1})}+\frac{C_{2}+k}{C_{1}+1}{\left[\sqrt{\frac{C_{2}+k}{c_{1}\left(C_{1}+1\right)}}\right]}^{\frac{1}{-(1+C_{1})}}.\; \end{array}$$Apparently, all coefficients involved in it can be determined.

### 3.2. QCD and EW Era

Based on the EoS outlined in Equation (9), and the dependence of energy density on the Hubble parameter, Equation (Equation 4), then Equation (Equation 11) can be reexpressed as
(21)$$\begin{array}{ccc} \ddot{a}\left(t\right)\;a\left(t\right)+C_{1}\;\dot{a}{\left(t\right)}^{2}+C_{2}\;a{\left(t\right)}^{2}+C_{1}\;k+4\pi \gamma_{2}a{\left(t\right)}^{2}\rho^{\delta_{2}} & = & 0, \end{array}$$
where $C_{1}$ and $C_{2}$ are variables depending on the corresponding EoS, Table 1. The last term in the lhs of this expression gives another difference with Equation (Equation 12). The other terms remaining can be estimated as outlined in the previous section. Now we focus on the the contributions added by this term,
(22)$$\begin{array}{c} 4\pi \gamma_{2}a{\left(t\right)}^{2}\rho^{\delta_{2}}=4\pi \gamma_{2}a{\left(t\right)}^{2(1-\delta_{2})}{\left(\frac{3}{8\pi k}\right)}^{\delta_{2}}{\left[1-\left(\frac{\Lambda a{\left(t\right)}^{2}}{3k}-\frac{\dot{a}{\left(t\right)}^{2}}{k}\right)\right]}^{\delta_{2}}. \end{array}$$

For $\delta_{2}=-0.576≃-0.5$, the square bracket can be expressed as a binomial expansion,
(23)$$\begin{array}{c} {\left[1-\left(\frac{\Lambda a{\left(t\right)}^{2}}{3k}-\frac{\dot{a}{\left(t\right)}^{2}}{k}\right)\right]}^{-1/2}=1-\frac{\Lambda a{\left(t\right)}^{2}}{6k}-\frac{\dot{a}{\left(t\right)}^{2}}{2k}+\cdots . \end{array}$$

Thus, we might approximate the entire bracket to first terms outlined. This result can also be obtained when assuming that the exponent $\delta_{2}$ approaches unity. Then, Equation (Equation 21) becomes
(24)$$\begin{array}{ccc} \ddot{a}\left(t\right)\;a\left(t\right)+C_{1}\dot{a}{\left(t\right)}^{2}+C_{2}a{\left(t\right)}^{2}+C_{1}k+C_{3}a{\left(t\right)}^{2}\left(1-C_{4}\frac{a{\left(t\right)}^{2}}{k}-\frac{\dot{a}{\left(t\right)}^{2}}{3k}\right) & = & 0. \end{array}$$

As done while solving Equation (Equation 15), we assume that $u\left(a\left(t\right)\right)=\dot{a}{\left(t\right)}^{2}$. Then, the approximated Equation (Equation 24) becomes
(25)$$\begin{array}{ccc} & & \frac{du(a(t))}{da(t)}+2C_{1}\frac{u(a(t))}{a(t)}+2C_{2}a\left(t\right)+2C_{1}k\;a{\left(t\right)}^{-1}+\\ & & 2C_{3}a\left(t\right)\left(1-C_{4}\frac{a{\left(t\right)}^{2}}{k}-\frac{u(a(t))}{3k}\right)=0,\; \end{array}$$
which can be solved as
(26)$$\begin{array}{ccc} u(t) & = & \dot{a}{\left(t\right)}^{2}\\ & = & \frac{1}{C_{3}}\left\{3k\left[C_{2}+C_{3}-3\left(1+C_{1}\right)C_{4}\right]-3C_{3}C_{4}a{\left(t\right)}^{2}+c_{1}a{\left(t\right)}^{-2C_{1}}e^{\frac{C_{3}}{3k}a{\left(t\right)}^{2}}\right.\\ & & \left.-C_{1}k\left[-3C_{2}-4C_{3}+9\left(1+C_{1}\right)C_{4}\right]e^{\frac{C_{3}}{3k}a{\left(t\right)}^{2}}Ein_{1-C_{1}}\left(\frac{C_{3}}{3k}a{\left(t\right)}^{2}\right)\right\},\; \end{array}$$
where $c_{1}$ is another integration constant to be fixed for boundary conditions and the exponential integral represents a special case of the incomplete gamma function
(27)$$\begin{array}{c} Ein_{n}\left(x\right)={\left(x\right)}^{n-1}\Gamma \left[1-n,y\right]. \end{array}$$

For Equation (Equation 26), there is no analytical solution. Nevertheless, the Hubble parameter, $H\left(t\right)=\dot{a}\left(t\right)/a\left(t\right)$, can be constructed as
(28)$$\begin{array}{ccc} H(t) & = & \frac{1}{C_{3}^{1/2}a\left(t\right)}\left\{3k\left[C_{2}+C_{3}-3\left(1+C_{1}\right)C_{4}\right]-3C_{3}C_{4}a{\left(t\right)}^{2}+c_{4}a{\left(t\right)}^{-2C_{1}}e^{\frac{C_{3}}{3k}a{\left(t\right)}^{2}}\right.\\ & & {\left.-C_{1}k\left[-3C_{2}-4C_{3}+9\left(1+C_{1}\right)C_{4}\right]e^{\frac{C_{3}}{3k}a{\left(t\right)}^{2}}Ein_{1-C_{1}}\left(\frac{C_{3}}{3k}a{\left(t\right)}^{2}\right)\right\}}^{1/2}. \end{array}$$

$C_{3}$ and $C_{4}$ are given in Table 1. The results obtained for the Hubble parameter as a function of the scale factor shall be presented.

### 3.3. Asymptotic Limit

Again, when substantiating Equations (Equation 4) and (Equation 10) into Equation (Equation 11), we get
(29)$$\begin{array}{ccc} \ddot{a}\left(t\right)\;a\left(t\right)+C_{1}\;\dot{a}{\left(t\right)}^{2}+C_{2}\;a{\left(t\right)}^{2}+C_{1}\;k & = & 0, \end{array}$$
which apparently looks almost identical to Equation (Equation 12) in Section 3.1. The *possible* analytical solution reads
(30)$$\begin{array}{ccc} u(t) & = & \dot{a}{\left(t\right)}^{2}\\ & = & c_{4}a{\left(t\right)}^{-2C_{1}}-\left[\frac{C_{2}}{C_{1}+1}+k\right]a{\left(t\right)}^{2}, \end{array}$$
for which the Hubble parameter can be given as
(31)$$\begin{array}{ccc} H(t) & = & {\left\{c_{4}a{\left(t\right)}^{-2C_{1}}-\left[\frac{C_{2}}{C_{1}+1}+k\right]a{\left(t\right)}^{2}\right\}}^{1/2}\frac{1}{a(t)}. \end{array}$$

By integrating (Equation 30), an expression for the scale factor can be obtained
(32)$$\begin{array}{ccc} a(t) & = & {\left[\sqrt{\frac{C_{2}+k\left(1+C_{1}\right)}{c_{1}\left(1+C_{1}\right)}}\right]}^{\frac{1}{-(1+C_{1})}}, \end{array}$$
which helps in constructing the corresponding Hubble parameter
(33)$$\begin{array}{ccc} H(t) & = & {\left[\frac{C_{2}+k\left(1+C_{1}\right)}{c_{1}\left(1+C_{1}\right)}\right]}^{\frac{1}{1+C_{1}}}\left\{-\frac{C_{2}+k\left(1+C_{1}\right)}{1+C_{1}}{\left[\sqrt{\frac{C_{2}+k\left(1+C_{1}\right)}{c_{1}\left(1+C_{1}\right)}}\right]}^{-\frac{2}{1+C_{1}}}\right.\\ & + & {\left.{\left[{\left[\sqrt{\frac{C_{2}+k\left(1+C_{1}\right)}{c_{1}\left(1+C_{1}\right)}}\right]}^{-\frac{1}{1+C_{1}}}\right]}^{-2C1}\right\}}^{1/2}. \end{array}$$

The various coefficients characterizing the various EoS and also combining cosmological constant, $C_{1}$, $C_{2}$, $C_{3}$, $C_{4}$, are listed in Table 1. Accordingly, analytical solutions similar to Equation (Equation 13) are obtained. The cosmological constant Λ is conjectured to count for the dark energy component. For Equation (Equation 29), expressions for Hubble parameter similar to (Equation 14) can then be derived. In addition, with the variable change $u=\dot{a}{\left(t\right)}^{2}$, (Equation 18) can be obtained, as well.

## 4. Cosmic Evolution in Viscous Approaches

### 4.1. Viscous Equations of State

The recent results for the bulk viscosity are based on non-perturbative and perturbative calculations with as much quark flavors as possible. By combining these calculations with additional dofs, such as photons, neutrinos, leptons, electroweak particles, and Higgs bosons, various thermodynamic quantities including bulk viscosity, for almost net-baryon-free cosmic matter, have been calculated up to the TeV-scale [16]. The dependence of the bulk viscosity on the energy density [26] is depicted in the top panel of Figure 2, in which both quantities are given in the physical units. As discussed, such a barotropic dependence straightforwardly allows for direct cosmological implications [19,20,21,24], where ρ(t) can be directly substituted by H(t), Equation (Equation 4). These wide values of ρ(t) which are accompanied by a wide range of temperatures cover quantum chromodynamic (QCD) (Hadron and QGP) and electroweak (EW) phases in the early Universe. Accordingly, the dependence of the bulk viscosity on the energy density can be parameterized.
(34)$$\begin{array}{ccc} \mathrm{Hadron}-\mathrm{QGP}: & & \zeta \left(t\right)=d_{1}+d_{2}\rho \left(t\right)+d_{3}\rho {\left(t\right)}^{d_{4}}, \end{array}$$
(35)$$\begin{array}{ccc} \mathrm{QCD}: & & \zeta \left(t\right)=e_{1}+e_{2}\rho {\left(t\right)}^{e_{3}}, \end{array}$$
(36)$$\begin{array}{ccc} \mathrm{EW}: & & \zeta \left(t\right)=f_{1}+f_{2}\rho {\left(t\right)}^{f_{3}}. \end{array}$$

The various fit parameters are given as follows. For Hadron-QCD: $d_{1}=-9.336\pm \;4.152$, $d_{2}=0.232\pm 0.003$, $d_{3}=11.962\pm 4.172$, and $d_{4}=0.087\pm 0.029$. For QCD: $e_{1}=8.042\pm 0.056$, $e_{2}=0.301\pm 0.002$, and $e_{3}=0.945\pm 0.0001$. For EW: $f_{1}=0.350\pm 0.065$, $f_{2}=10.019\pm 0.934$, and $f_{3}=0.929\pm 8.898\times 10^{-5}$.

While ζ(t) vs. ρ(t) is very structured in the hadron era, there are three domains to be emphasized (from low to large energy density).

The first one is the hadron-QGP domain (Hadron-QGP), which spans over ρ(t)⪅100GeV/fm${}^{3}$. At the beginning, there is a rapid increase in ζ(t), i.e., ζ≂1GeV${}^{3}$, at ρ(t)≃1GeV/fm${}^{3}$, which is then followed by a slight increase in ζ(t). For example, at ρ(t)≃100GeV/fm${}^{3}$, zeta(t) reaches ∼130GeV${}^{3}$. It is apparent that the hadron–parton phase transition seems to take place at ρ(t)⪅0.5GeV/fm${}^{3}$[34,35]. At this value, ζ(t)⪅0.5GeV${}^{3}$.The second domain, the QGP epoch, seems to be formed, at 0.5⪅ρ(t)⪅100GeV/fm${}^{3}$, i.e., a much wider ρ(t) than that of the hadron domain. Thus, we could conclude that over this wide range of ρ(t), the bulk viscosity is obviously not only finite but rather largely supporting the RHIC discovery of strongly correlated *viscous* QGP [3,4,6]. At higher ρ(t), we observe a tendency of a linear increase in ζ(t) with further increasing ρ(t). Thus, the second domain is the one where $100⪆\rho \left(t\right)⪅5\times 10^{7}\;$GeV/fm${}^{3}$ and $80⪆\zeta \left(t\right)⪅10^{6}\;$GeV${}^{3}$. In light of this observation, we conclude that the phase transition from QCD to EW domain is very smooth.The third domain is also characterized by an almost linear increase in ζ(t) with increasing ρ(t). For $10^{8}⪅\rho \left(t\right)⪅10^{15}\;$GeV/fm${}^{3}$, there is a nearly steady increase in ζ(t) from $10^{8}$ to $10^{14}\;$GeV${}^{3}$.

For temperatures ranging from a few MeV to TeV and energy densities up to $10^{16}\;$GeV/fm${}^{3}$, we have taken into consideration almost all possible contributions to the bulk viscosity. With these we mean the thermodynamic quantities calculated in non-perturbation and perturbation QCD with up, down, strange, charm, and bottom quark flavors. The second type of contributions is the gauge bosons, the entire gluonic sector. We have also included photons, $W^{\pm }$, and $Z^{0}$, charged leptons (neutrino, electron, muon, and tau), and scalar Higgs particle. The third type of contributions is the vacuum and thermal condensations. We have included condensations for up, down, strange, and charm quarks. We merely still miss the vacuum and the thermal bottom quark condensates, besides the entire gravitational, the neutral leptons, and the top quark sector to compile the entire standard model for elementary particles.

The dependence of temperature T(t) on the energy density ρ(t), the barotropic equation of state, is depicted in bottom panel of Figure 2. We notice that T(t) almost linearly depends on ρ(t). A best parameterization reads
(37)$$\begin{array}{c} T\left(t\right)=\alpha_{4}+\beta_{4}\rho {\left(t\right)}^{\gamma_{4}}, \end{array}$$
where $\alpha_{4}=0.048\pm 0.001$, $\beta_{4}=0.13\pm 2\times 10^{-4}$, and $\gamma_{4}=0.25\pm 8\times 10^{-5}$. We notice that at low ρ(t) the temperature looks a little bit structured. While with increasing ρ(t), the temperature goes almost linearly with increasing ρ(t), especially at very high temperatures, where ρ(t) becomes related in $T^{4}$, i.e., ideal gas.

The third quantity, for which we need to propose a barotropic EoS, is the relaxation time, τ(t). We assume to apply the phenomenological model presented in refs. [18,19,24,36], which is based on dissipative relativistic fluid. This model was assumed to characterize the evolution of the Universe with a flat homogeneous isotropic Friedmann–Robertson–Walker geometry filled with viscous cosmic fluid, but still valid for other types of curvature and cosmic backgrounds. Accordingly, we have
(38)$$\begin{array}{ccc} \tau (t) & = & \frac{\zeta (t)}{\rho (t)}. \end{array}$$

Figure 3 shows the energy–density dependence of the relaxation time τ(t). Bearing in mind the linear dependence of the energy density ρ(t) on the temperature $T^{4}$, bottom panel of Figure 2 and Equation (Equation 37), the temperature dependence τ(t) can be almost straightforwardly estimated. As done in the present script, we would like to distinguish between hadron-QGP (squares), QCD (circles), and electroweak eras (diamonds). In the hadron-QGP era, there is a very rapid decrease in τ(t) with increasing rho(t). The QCD epoch is characterized by a slower decline in τ(t) with increasing rho(t). The relaxation time within the electroweak epoch starts and ends with a slow decrease, while in the middle EW era, τ(t) rapidly decreases with increasing ρ(t). This region likely characterizes the electroweak phase transition. In this case, τ(t) is conjectured to play the role of an order parameter. For the electroweak phase transition, other thermodynamic order parameters should be proposed and then analyzed. The dependence of τ(t) on rho(t) is proposed as follows.
(39)Hadron−QGP:τ(t)=g1+g2exp(−g3ρ(t)g4),(40)QCD:τ(t)=h1+h2h3+log(h4ρ(t)),(41)EW:τ(t)=k1ρ(t)k2log(k3ρ(t)),
where $g_{1}=0.0002\pm 1.945\times 10^{-5}$, $g_{2}=0.008\pm 0.001$, $g_{3}=1.671\pm 0.097$, $g_{4}=0.312\pm 0.0226$, $h_{1}=-1.605\times 10^{-7}\pm 2.556\times 10^{-6}$, $h_{2}=0.0015\pm 0.0013$, $h_{3}=0.935\pm 1.084\times 10^{5}$, $h_{3}=10.524\pm 1.141\times 10^{6}$, $k_{1}=9.582\times 10^{-4}\pm 9.504\times 10^{-5}$, $k_{2}=0.216\pm 0.0035$, and $k_{3}\;=\;5.9\times 10^{-7}\pm 9.165\times 10^{-8}$.

In the sections that follow, we apply well-know theories for relativistic dissipative fluid to the cosmic background. We start with the Eckart relativistic theory of a simple dissipative fluid, which is used to simplify the nuclear motion arising in the second Born–Oppenheimer approximation. The cosmic relevance of this theory is remarkable because it introduces the so-called Eckart frame, which is a frame of orthonormal vectors following a vibro-rotating object. The orientation of this frame is governed by the so-called Eckart conditions assuring minimal Coriolis interaction. Second, we apply the Israel–Stewart theory as this theory is conjectured to solve Eckart theory’s lack of causality and its obvious instabilities by introducing a second-order term to the entropy.

### 4.2. Eckart Relativistic Viscous Fluid

For cosmological context, the first theory of relativistic dissipative fluid has been presented by Eckart [37] and Landau and Lifshitz [38]. It was pointed out that regardless the choice of EoS, the equilibrium states of this theory are found unstable [39]. In this theory, only the first-order deviation from the equilibrium is taken into consideration. This leads to the superluminal velocities of the dissipative signals, i.e., signals propagate through the relativistic dissipative fluid with velocities exceeding the speed of light *c* and hence the theory violates the causality principle [40]. Moreover, it was shown that the resulting equilibrium states are unstable [41]. All these severe problems are originated from the fact that the Eckart theory merely considers first-order deviations from the equilibrium leading to parabolic differential equations, Equation (Equation 45). The applicability of this theory can only be thought for quasi-stationary phenomena, i.e., temporally and spatially slowly varying, which are characterized by mean free-path and mean collision-time.

The Eckart theory introduces a linear relationship between the bulk viscous pressure and the rate of expansion of the Universe [42]. Obviously, this feature—despite the severe contents—makes it possible to work out an analytical method for the cosmic parameters in the expanding Universe. For bulk viscous cosmic fluid, whose energy–momentum tensors are given as
(42)$$\begin{array}{ccc} T^{\mu \nu } & = & \left(\rho +p+\Pi \right)\;u_{\mu }\;u^{\nu }-\left(p+\Pi \right)\delta_{\mu }^{\nu }, \end{array}$$
the line element in flat homogeneous isotropic Friedmann–Robertson–Walter metric reads
(43)$$\begin{array}{ccc} ds^{2} & = & dt^{2}+a{\left(t\right)}^{2}\left[dx^{2}+dy^{2}+dz^{2}\right]. \end{array}$$

When applying Eckart theory on modeling such a cosmic fluid, we assume averaged 4-velocity fields $u^{\alpha }$ with $u^{\alpha }u_{\alpha }=1$ and vector number density $n^{\alpha }=n\;u^{\alpha }$. For unbalanced creation/annihilation processes in gravitational fields, $n_{;\alpha }^{\alpha }=0$,
(44)$$\dot{n}+3\;H\;n=0,$$
with Hubble parameter $H=u_{;\alpha }^{\alpha }$. In this theory, the entropy current is given as
(45)$$\begin{array}{ccc} S^{\alpha } & = & s\;n\;u^{\alpha }. \end{array}$$

As discussed, this is a non-conserved quantity. The covariant form of second law of thermodynamics reads $S_{;\alpha }^{\alpha }\geq 0$. The divergence of this quantity is given as $TS_{;\alpha }^{\alpha }=-3H\Pi$.

With respect to the proposed cosmic fluid, the temporal evolution can be related to the dynamical laws of the particle number conservation $N_{;i}^{i}=0$. Gibbs equation implies that $Td\rho =d\left(\rho /n\right)+pd\left(1/n\right)$. Then, the covariant entropy current assures a linear *first-order* relationship between the thermodynamical flux Π(t) and the corresponding H(t). It is worth highlighting that H(t) in this context plays the role of a force.
(46)Π(t)=−3ζ(t)H(t).

Having an estimation of the bulk viscous pressure Π, we can now substitute Equation (Equation 46) in Equation (Equation 6),
(47)$$\begin{array}{ccc} \dot{H}\left(t\right) & = & -4\;\pi \;\left[\rho (t)+\;p(t)-3\;\zeta (t)\;H(t)\right]+\frac{k}{a{\left(t\right)}^{2}}, \end{array}$$
which can be rewritten as
(48)$$\begin{array}{ccc} \ddot{a}\left(t\right)\;a\left(t\right)-\left[1+12\pi \zeta (t)\right]\dot{a}{\left(t\right)}^{2}+4\pi \;\left[\rho (t)+p(t)\right]a{\left(t\right)}^{2}-k & = & 0. \end{array}$$

For the present calculations, we start with Equation (Equation 48), in which we substitute with the barotropic EoS of the pressure and the bulk viscosity. They can then be related to the Hubble parameter, Equation (Equation 4). Due to the various barotropic EoS introduced, Equations (Equation 8)–(Equation 10), and Section 4.1, the evolution of the various cosmological parameters obviously differ from epoch to epoch. Equation (Equation 48) combines extended assumptions and ingredients of SMC. This is viscous cosmic background.

The sections that follow elaborate details on the dependence of H(t) on a(t), where both quantities are functions of the cosmic time *t*. Such a limitation is merely based on the currently available analytical solutions. When lifting such mathematical limitations, bSMC emerges as a proper cosmological approach.

#### 4.2.1. Hadron-QGP Era

Substituting with the pressure, Equation (Equation 8), and the bulk viscosity, Equation (Equation 34), into Equation (Equation 48) leads to a second-order differential equation
(49)$$\begin{array}{ccc} \ddot{a}\left(t\right)a\left(t\right)-D_{1}\dot{a}{\left(t\right)}^{2}+D_{2}a{\left(t\right)}^{2}+D_{3}k & - & \\ D_{4}\left[1-D_{5}a{\left(t\right)}^{2}-\frac{\dot{a}{\left(t\right)}^{2}}{2k}\right]-\frac{9}{2}d_{2}a{\left(t\right)}^{-2}\dot{a}{\left(t\right)}^{4}-\frac{9}{2}d_{2}a{\left(t\right)}^{-2}\dot{a}{\left(t\right)}^{2} & = & 0, \end{array}$$
where the various coefficients of the different EoS are included in the following parameters
$$\begin{array}{ccc} D_{1} & = & 1+12\pi d_{1}-\frac{3}{2}\left(1+\beta_{1}\right)-\frac{3}{2}d_{2}\Lambda ,\\ D_{2} & = & 4\pi \alpha_{1}-\frac{\Lambda }{2}\left(1+\beta_{1}\right),\\ D_{3} & = & \frac{3}{2}\left(1+\beta_{1}\right)-1,\\ D_{4} & = & 2^{2-3d_{4}}3^{1+d_{4}}\pi^{1-d_{4}}d_{3}k^{-d_{4}},\\ D_{5} & = & \frac{\Lambda }{6k}. \end{array}$$

For Equation (Equation 49), there is no analytical solution. If assuming that $u\left(a\left(t\right)\right)=\dot{a}{\left(t\right)}^{2}$, this can be reduced to
(50)$$\begin{array}{ccc} u^{{}^{\prime }}\left(a\left(t\right)\right)-2D_{1}\frac{u(a(t))}{a(t)}+2D_{2}a\left(t\right)+2D_{3}ka{\left(t\right)}^{-1} & - & \\ 2D_{4}{\left[1+\left(\frac{u(a(t))}{k}-\frac{\Lambda }{3k}\right)\right]}^{d_{4}}a{\left(t\right)}^{-1}-9d_{2}u{\left(a\left(t\right)\right)}^{2}a{\left(t\right)}^{-3}-9d_{2}ku\left(a\left(t\right)\right)a{\left(t\right)}^{-3} & = & 0,\; \end{array}$$
where $u^{{}^{\prime }}\left(a\left(t\right)\right)=du\left(a\left(t\right)\right)/da\left(t\right)$. With binomial expansion, the squared bracket can be approximated to unity.

This reduces the certainty of the proposed solution but this seems to be the only approximation possible! T.

Such an assumption, which, on one hand, reduces the certainty of the proposed solution but this seems to be the only approximation possible but on other hand the inclusion of higher terms simply prevents any analytical solution, leads to an analytical solution,
(51)$$\begin{array}{ccc} u(a(t)) & = & \dot{a}{\left(t\right)}^{2}\\ & = & -\frac{3^{\frac{A}{k}}{\left(d_{2}k\right)}^{\frac{A}{2k}}}{K_{1}\left(t\right)}\left\{\frac{18d_{2}k^{2}M\left[\frac{5D_{4}-2k\left(D_{1}+2D_{3}-3\right)+A}{4k},2+\frac{A}{2k},-\frac{9kd_{2}}{2a{\left(t\right)}^{2}}\right]}{{\left[5D_{4}-2k\left(D_{1}+2D_{3}-1\right)+A\right]}^{-1}\left(2k+A\right)}\right\}a{\left(t\right)}^{-\frac{A}{k}}\\ & + & \frac{2\;3^{\frac{A}{k}}{\left(d_{2}k\right)}^{\frac{A}{2k}}}{K_{1}\left(t\right)}\left\{\left(D_{4}+2k-2kD_{1}+A\right)M\left[\frac{5D_{4}-2k\left(D_{1}+2D_{3}-1\right)+A}{4k},1+\frac{A}{2k},-\frac{9kd_{2}}{2a{\left(t\right)}^{2}}\right]\right\}a{\left(t\right)}^{2-\frac{A}{k}}\\ & - & \frac{c_{4}2^{\frac{A}{2k}}}{K_{1}\left(t\right)}\left\{\frac{18d_{2}k^{2}M\left[-\frac{-5D_{4}+2k\left(D_{1}+2D_{3}-3\right)+A}{4k},2-\frac{A}{2k},-\frac{9kd_{2}}{2a{\left(t\right)}^{2}}\right]}{{\left[-5D_{4}+2k\left(D_{1}+2D_{3}-1\right)+A\right]}^{-1}\left(A-2k\right)}\right\}\\ & - & \frac{2c_{4}2^{\frac{A}{2k}}}{K_{1}\left(t\right)}\left[2k\left(D_{1}-1\right)-D_{4}+A\right]M\left[-\frac{-5D_{4}+2k\left(D_{1}+2D_{3}-1\right)+A}{4k},1-\frac{A}{2k},-\frac{9kd_{2}}{2a{\left(t\right)}^{2}}\right]a{\left(t\right)}^{2}, \end{array}$$
where
(52)$$\begin{array}{ccc} K_{1}\left(t\right) & = & 36d_{2}k\left\{3^{\frac{A}{k}}{\left(d_{2}k\right)}^{\frac{A}{2k}}M\left[\frac{1}{4k}\left(5D_{4}-2k\left(D_{1}+2D_{3}-1\right)+A\right),1+\frac{A}{2k},-\frac{9kd_{2}}{2a{\left(t\right)}^{2}}\right]a{\left(t\right)}^{-\frac{A}{k}}\right.\\ & & \left.\;+c_{4}2^{\frac{A}{2k}}M\left[\frac{-1}{4k}\left(-5D_{4}+2k\left(D_{1}+2D_{3}-1\right)+A\right),1-\frac{A}{2k},-\frac{9kd_{2}}{2a{\left(t\right)}^{2}}\right]\right\},\\ A & = & {\left[72d_{2}\left(D_{2}+D_{4}D_{5}\right)k^{2}+{\left[D_{4}-2\left(D_{1}-1\right)k\right]}^{2}\right]}^{1/2} \end{array}$$

Even for the resultant differential Equation (Equation 51) there is no analytical solution in terms of the cosmic time *t*. The only possible solution is $\dot{a}\left(t\right)$ as a function of a(t). Such a solution (impeded in Equation (Equation 53)) leads to an analytical expression for H(t) as a function of a(t), i.e., functionality, which in turn depends on regularized confluent hypergeometric function whose asymptotic limit reads $M\left(\left\{a,b,z\right\}\right)∼\Gamma \left(b\right)\left(e^{z}z^{a-b}+{\left(-\right)}^{-a}/\Gamma \left(b-a\right)\right)$ [43]. Two of the three regular singularities of M({a,b,z}) are conjectured to merge into an irregular singularity and therefrom the conjugate “confluent” emerges. The Hubble parameter reads
(53)$$\begin{array}{ccc} H(t) & = & \left\{-\frac{3^{\frac{A}{k}}{\left(d_{2}k\right)}^{\frac{A}{2k}}}{K_{1}\left(t\right)}\left[\frac{18d_{2}k^{2}M\left[\frac{5D_{4}-2k\left(D_{1}+2D_{3}-3\right)+A}{4k},2+\frac{A}{2k},-\frac{9kd_{2}}{2a{\left(t\right)}^{2}}\right]}{{\left[5D_{4}-2k\left(D_{1}+2D_{3}-1\right)+A\right]}^{-1}\left(2k+A\right)}\right]a{\left(t\right)}^{-2-\frac{A}{k}}\right.\\ & + & \left.\frac{2\;3^{\frac{A}{k}}{\left(d_{2}k\right)}^{\frac{A}{2k}}}{K_{1}\left(t\right)}\left[\left(D_{4}+2k-2kD_{1}+A\right)M\left[\frac{5D_{4}-2k\left(D_{1}+2D_{3}-1\right)+A}{4k},1+\frac{A}{2k},-\frac{9kd_{2}}{2a{\left(t\right)}^{2}}\right]\right]a{\left(t\right)}^{-\frac{A}{k}}\right.\\ & - & \left.\frac{c_{4}2^{\frac{A}{2k}}}{K_{1}\left(t\right)}\left[\frac{18d_{2}k^{2}M\left[-\frac{-5D_{4}+2k\left(D_{1}+2D_{3}-3\right)+A}{4k},2-\frac{A}{2k},-\frac{9kd_{2}}{2a{\left(t\right)}^{2}}\right]}{{\left[-5D_{4}+2k\left(D_{1}+2D_{3}-1\right)+A\right]}^{-1}\left(A-2k\right)}\right]a{\left(t\right)}^{-2}\right.\\ & - & {\left.\frac{2c_{4}2^{\frac{A}{2k}}}{K_{1}\left(t\right)}\left[2k\left(D_{1}-1\right)-D_{4}+A\right]M\left[-\frac{-5D_{4}+2k\left(D_{1}+2D_{3}-1\right)+A}{4k},1-\frac{A}{2k},-\frac{9kd_{2}}{2a{\left(t\right)}^{2}}\right]\right\}}^{1/2}. \end{array}$$

It is worth highlighting that Kummer confluent hypergeometric functions, for instance, which are common standard forms of the confluent hypergeometric functions M, have a regular singular point, at $z\equiv -9kd_{2}/2a{\left(t\right)}^{2}=0$ and an irregular singular point at $z\equiv -9kd_{2}/2a{\left(t\right)}^{2}=\infty$. Thus, the curvature parameter *k* and the scale factor a(t) define whether a regular or irregular singular point appears. At vanishing and finite cosmological constants, the numerical results of H(t) vs. a(t) shall be presented.

#### 4.2.2. QCD-EW Era

When substituting with the barotropic equation for the pressure, Equation (9), and the bulk viscosity, Equation (35), in Equation (Equation 48), we get
(54)$$\begin{array}{ccc} & & \ddot{a}\left(t\right)a\left(t\right)-E_{1}\dot{a}{\left(t\right)}^{2}+E_{2}a{\left(t\right)}^{2}+E_{3}k-E_{4}{\left[1+\left(\frac{\dot{a}{\left(t\right)}^{2}}{k}-\frac{\Lambda }{3k}\right)\right]}^{e_{3}}\frac{\dot{a}{\left(t\right)}^{2}}{a{\left(t\right)}^{2e_{3}}}+\\ & & E_{5}{\left[1+\left(\frac{\dot{a}{\left(t\right)}^{2}}{k}-\frac{\Lambda }{3k}\right)\right]}^{\delta_{2}}a{\left(t\right)}^{2-2e_{3}}=0, \end{array}$$
with the coefficients
$$\begin{array}{ccc} E_{1} & = & 1+12\pi e_{1}-\frac{3}{2}\left(1+\beta_{2}\right),\\ E_{2} & = & 4\pi \alpha_{2}-\frac{\Lambda }{2}\left(1+\beta_{2}\right),\\ E_{3} & = & \frac{3}{2}\left(1+\beta_{2}\right)-1,\\ E_{4} & = & 2^{2-3e_{3}}3^{1+e_{3}}\pi^{1-e_{3}}e_{2}k^{-e_{3}},\\ E_{5} & = & 2^{2-3\delta_{2}}3^{\delta_{2}}\pi^{1-\delta_{2}}\gamma_{2}k^{-\delta_{2}}. \end{array}$$

Assuming that $u\left(a\left(t\right)\right)={\dot{a}}^{2}\left(t\right)$ and applying the same approximation given in Equation (Equation 23), the previous differential equation can be reduced to
(55)$$\begin{array}{ccc} u^{\prime }\left(a\left(t\right)\right)-2E_{1}\frac{u(a(t))}{a(t)}+2E_{2}a\left(t\right)-2k\frac{E_{3}}{a(t)} & + & \\ 2E_{4}\left[1-C_{4}\frac{a{\left(t\right)}^{2}}{k}-\frac{u(a(t))}{3k}\right]\frac{u(a(t))}{a{\left(t\right)}^{1+2e_{3}}}+2E_{5}\left[1-C_{4}\frac{a{\left(t\right)}^{2}}{k}-\frac{u(a(t))}{3k}\right]a{\left(t\right)}^{1-3\delta_{3}} & = & 0.\; \end{array}$$

An analytical solution is only possible when both squared brackets are replaced by unity
(56)$$\begin{array}{ccc} u(a(t)) & = & \dot{a}{\left(t\right)}^{2}\\ & = & \left\{\left[c_{4}\;e_{3}\;a{\left(t\right)}^{2E_{1}}-E_{3}\;k\;L_{1-\frac{E_{1}}{e_{3}}}\left(-\frac{E_{4}}{e_{3}}a{\left(t\right)}^{-2e_{3}}\right)\right.\right.\\ & - & \left.\left.E_{2}\;L_{\frac{1-E_{1}+e_{3}}{e_{3}}}\left(-\frac{E_{4}}{e_{3}}a{\left(t\right)}^{2}\right)a{\left(t\right)}^{2}\right]a{\left(t\right)}^{2\delta_{2}}\right.\\ & - & \left.E_{5}\;L_{1-\frac{E_{1}-1+\frac{3}{2}\delta_{2}}{e_{3}}}\left(-\frac{E_{4}}{e_{3}}a{\left(t\right)}^{-2e_{3}}\right)a{\left(t\right)}^{2}\right\}\frac{e^{-\frac{E_{4}}{e_{3}}a{\left(t\right)}^{-2e_{3}}}}{e_{3}}a{\left(t\right)}^{-2\delta_{2}}, \end{array}$$
where
(57)$$\begin{array}{ccc} L_{\nu }\left(z\right) & = & \int_{1}^{\infty }\frac{e^{-zt}}{t^{\nu }}dt. \end{array}$$

Thus, the corresponding Hubble parameter reads
(58)$$\begin{array}{ccc} H(t) & = & \frac{1}{a(t)}\left\{\left[\left[c_{4}e_{3}a{\left(t\right)}^{2E_{1}}-E_{3}k\;L_{1-\frac{E_{1}}{e_{3}}}\left(-\frac{E_{4}}{e_{3}}a{\left(t\right)}^{-2e_{3}}\right)\right.\right.\right.\\ & - & \left.\left.\left.E_{2}\;L_{\frac{1-E_{1}+e_{3}}{e_{3}}}\left(-\frac{E_{4}}{e_{3}}a{\left(t\right)}^{2}\right)a{\left(t\right)}^{2}\right]a{\left(t\right)}^{2\delta_{2}}\right.\right.\\ & - & {\left.\left.E_{5}\;L_{1-\frac{E_{1}-1+\delta_{2}}{e_{3}}}\left(-\frac{E_{4}}{e_{3}}a{\left(t\right)}^{-2e_{3}}\right)a{\left(t\right)}^{2}\right]\frac{e^{-\frac{E_{4}}{e_{3}}a{\left(t\right)}^{-2e_{3}}}}{e_{3}}\;a{\left(t\right)}^{-2\delta_{2}}\right\}}^{1/2}. \end{array}$$

#### 4.2.3. EW (Asymptotic) Era

For pressure, Equation (Equation 10), and bulk viscosity, Equations (36) and (Equation 48) leads to
(59)$$\begin{array}{ccc} \ddot{a}\left(t\right)a\left(t\right)-F_{1}{\dot{a}}^{2}\left(t\right)-F_{2}a^{2}\left(t\right)+F_{3}k-F_{4}{\left[1\left(\frac{{\dot{a}}^{2}\left(t\right)}{k}-\frac{\Lambda }{3k}a^{2}\left(t\right)\right)\right]}^{f_{3}}a^{-2f_{3}}\left(t\right)\;{\dot{a}}^{2}\left(t\right) & = & 0,\; \end{array}$$
where
$$\begin{array}{ccc} F_{1} & = & 1+12\pi f_{1}-\frac{3}{2}\left(1+\gamma_{3}\right),\\ F_{2} & = & \frac{\Lambda }{2}\left(1+\gamma_{3}\right),\\ F_{3} & = & \frac{3}{2}\left(1+\gamma_{3}\right)-1,\\ F_{4} & = & 2^{2-3f_{3}}3^{1+f_{3}}\pi^{1-f_{3}}f_{2}k^{-f_{3}}. \end{array}$$

Applying the same substitution, $u\left(a\left(t\right)\right)=\dot{a}{\left(t\right)}^{2}$, and taking into account the first term of the binomial expansion as unity, we get an analytical solution, functionality,
(60)$$\begin{array}{ccc} u(a(t)) & = & \dot{a}{\left(t\right)}^{2}\\ & = & \frac{e^{-\frac{F_{4}}{f_{3}}a{\left(t\right)}^{-2f_{3}}}}{f_{3}}\left\{c_{4}f_{3}a{\left(t\right)}^{2F_{1}}-F_{3}kL_{1-\frac{F_{1}}{f_{3}}}\left(-\frac{F_{4}}{f_{3}}a{\left(t\right)}^{-2f_{3}}\right)\right.\\ & + & \left.F_{2}L_{\frac{1-F_{1}+f_{3}}{f_{3}}}\left(-\frac{F_{4}}{f_{3}}a{\left(t\right)}^{-2f_{3}}\right)a{\left(t\right)}^{2}\right\}.\; \end{array}$$

Accordingly, the Hubble parameters is given as
(61)$$\begin{array}{ccc} H(t) & = & \frac{1}{a(t)}\left[\frac{e^{-\frac{F_{4}}{f_{3}}a{\left(t\right)}^{-2f_{3}}}}{f_{3}}\left\{c_{4}f_{3}a{\left(t\right)}^{2F_{1}}-F_{3}kL_{1-\frac{F_{1}}{f_{3}}}\left(-\frac{F_{4}}{f_{3}}a{\left(t\right)}^{-2f_{3}}\right)\right.\right.\\ & + & {\left.\left.F_{2}L_{\frac{1-F_{1}+f_{3}}{f_{3}}}\left(-\frac{F_{4}}{f_{3}}a{\left(t\right)}^{-2f_{3}}\right)a{\left(t\right)}^{2}\right\}\right]}^{1/2}.\; \end{array}$$

### 4.3. Israel–Stewart Relativistic Viscous Fluid

In order to fix the acausality and instability problem of Eckart theory, Israel and Stewart have introduced a relativistic second-order theory for relativistic fluid [44,45]. With *extended* irreversible thermodynamics, this theory was then developed by Hiscock and Lindblom [46]. This theory is also characterized by a deviation from equilibrium as defined by Eckart theory. Quantities such as bulk stress, heat flow, and shear stress are treated as independent dynamical variables. Accordingly, 14 dynamical fluid variables have to be estimated. The role that this type of causal thermodynamics would play in the general theory of relativity was reported in ref. [18]. A general *algebraic* form for $S^{\alpha }$ including a *second-order* term in the dissipative thermodynamical flux Π [44,45,47] reads
(62)$$S^{\alpha }=s\;n\;u^{\alpha }+\frac{\tau }{\zeta }\;\Pi^{2}\;\frac{u^{\alpha }}{2T},$$
where τ is the relaxation time. Similar to Eckart theory, the corresponding number flux could be given as
(63)$$\begin{array}{ccc} N^{\alpha } & = & N\;u^{\alpha }. \end{array}$$

For the evolution of the bulk viscous pressure, we adopt the causal evolution equation in the simplest way, i.e., linear in Π satisfying the *H*-theorem [18]. Accordingly, the entropy production remains nonnegative, $S_{;i}^{i}=\Pi^{2}/\zeta T\geq 0$ [44,45]. The causal transport equation of the bulk viscous pressure reads [18]
(64)$$\tau \dot{\Pi }+\Pi =-3\;\zeta \;H-\frac{\epsilon }{2}\;\tau \;\Pi \;\left(3\;H+\frac{\dot{\tau }}{\tau }-\frac{\dot{\zeta }}{\zeta }-\frac{\dot{T}}{T}\right),$$
where ϵ is a parameter controlling the type of considered theory. ϵ=1 assures full theory, while ϵ=0 a truncated one. It is obvious that the non-causal Eckart theory can be retrieved, Equation (Equation 46), at τ=0. In order to have a closed system from Equations (Equation 4), (Equation 7), and (Equation 64), we have to introduce EoS for the pressure p(t), the temperature T(t), bulk viscosity coefficient ζ(t), and the relaxation time τ(t), respectively, Section 4.1.

In the sections that follow, we elaborate the consequences of the various barotropic EoS for p(t), T(t), ζ(t), and τ(t) in strong of electroweak epochs of the early Universe. We get sophisticated differential equations. We hope that this concrete mathematical problem finds resonances among mathematicians. Despite their apparent challenging analytical solutions, we separately derive them in the Appendix A. A future work shall be devoted in order to propose numerical solutions for all these differential equations.

## 5. Results

The present section summarizes the results of the possible analytical solutions outlined in Section 3 and Section 4.2. They are only limited to the Hubble parameter in dependence on the scale factor for non-viscous, Section 3, and Eckart-type viscous cosmic backgrounds, Section 4.2. As introduced, from the corresponding EoS, we could differentiate between the various epochs of the early Universe. Nevertheless, we did not emphasize when each epoch starts or when ends, i.e., in terms of the cosmic time. Such a concrete limitation becomes only possible when the initial and the final conditions are precisely determined. This is not precisely available. As alternatives, we would be able to propose for each epoch an interval of cosmic energy densities, which in turn could be related to an interval of the Hubble parameter. The latter can then be given as functions of the scale factor the proposed analytical solutions. On the other hand, such a concrete limitation would be only urgently needed, when a complete or an inter-epochal picture is to be drawn. The results discussed in the sections that follow are not limiting the evolution of the Hubble parameter within the successive epochs. They cover a wider range than that of the corresponding epoch. Accordingly, we conclude that the evolution during the successive epochs characterized by electroweak and strong interactions would not be monotonic.

### 5.1. Non-Viscous Fluid

Figure 4 depicts the dependence of the Hubble parameter on the scale factor at finite (top panel) and vanishing cosmological constant (top panel). The results for the equations of state characterizing hadron, QCD-EW, and asymptotic limit are presented as dashed, dotted, long dashed curves, respectively. We also draw the ideal gas results as tiny dashed curves. There is a rapid decrease in H(t) with increasing a(t). The various epochs (the different EoS) show miscellaneous rates. Relative to the ideal gas EoS, hadron, and asymptotic EoS look very similar, especially, at finite cosmological constant (top panel). At large a(t), both hadron and asymptotic EoS become almost identical. The QCD/EW EoS shows a slightly different behavior, especially at large a(t), where H(t) diminishes.

At vanishing cosmological constant (bottom panel), the rate of decreasing H(t) with the increase in a(t) is larger than the one observed in the top panel. Here, only QCD/EW EoS looks similar to the ideal gas EoS, while both hadron and asymptotic epochs look almost identical. At small a(t), both have a similar decrease as the one of ideal and QCD/EW EoS, while, at large a(t), their corresponding H(t) vanishes.

As discussed, each EoS should be restrictively utilized within a specific interval of the cosmic time characterizing the corresponding cosmic epoch. Due to the mathematical difficulties associated with the resulting differential equations so that the proposed analytical solutions are restricted to the functionality H(a(t)) but not in terms of the cosmic time *t*, directly, we are left with a unique alternative. This is relating the various epochs to an interval of energy densities, as introduced in ref. [16,26] and Section 4.1. It is obvious that even this option is an approximation. Thus, we leave the results drawn in Figure 4 unchanged. The conclusion which could be drawn here is that the cosmic evolution (H(t) vs. a(t)) seems not monotonic, especially over the successive asymptotic, EW-QCD, and hadron epochs.

### 5.2. Eckart-Type Viscous Fluid

Figure 5 presents the same as in in Figure 4 but here for viscous cosmic geometry (Eckart theory). Comparing with the results depicted in Figure 4, the dependence of H(t) vs. a(t) in viscous background geometry looks very different. Hadronic EoS is associated with non-singularity. Finite cosmological constant likely assures non-singularity in both quantities, while vanishing cosmological constant is associated with non-singular Hubble parameter. The QCD-EW EoS results in diverging Hubble parameter within a tiny range of the scale parameter. At lower a(t), we find that H(t) remains almost vanishing. For the asymptotic EoS, at finite cosmological constant and low a(t), H(t) vanishes. Then, H(t) gets positive small values. At higher a(t), H(t) becomes non-physical. Again, within the short range of a(t), H(t) diverges. For the asymptotic EoS, at vanishing cosmological constant and low a(t), H(t) vanishes. Then increasing a(t), the resulting H(t) very slightly linearly decreases. Then, H(t) diverges within the short range of a(t).

## 6. Conclusions

Based on recent progress achieved, especially the numerical and experimental studies on hadron, parton, and EW matter, the main conclusion of the present study is that the analytical solutions for EoS, in which as many as possible contributions from both standard model for elementary particles and standard model for cosmology are taken into consideration, are sophisticated. The only possible analytical solutions are the ones relating the Hubble parameter to the scale factor, functionality, in non-viscous and Eckart-type-viscous cosmic backgrounds. For Israel–Stewart-viscosity, the resulting differential equations are challenging tasks for mathematicians. We have outlined these differential equations as road-maps for future studies.

Recent non-perturbative and perturbative calculations with as many as possible quark flavors at almost physical masses have been combined with the thermal contributions from photons, charged neutrinos, leptons, electroweak particles ($W^{\pm }$ and $Z^{0}$ bosons), and the scalar Higgs bosons. Various thermodynamic quantities, including pressure, energy density, bulk viscosity, relaxation time, and temperature for almost net-baryon-free cosmic matter could be calculated up to the TeV-scale, i.e., covering hadron, QGP, and electroweak (EW) phases.

In equivalence with Newtonian mechanics and based on Friedman solutions and the conservation of the energy–momentum tensor, McCrea and Milde and Peebles derived the temporal evolution of the energy density, i.e., an equation of motion, with vanishing and finite pressure, that dictates that the decrease in the energy content of the Universe is given by the energy budget due to the expansion and the work done by the pressure. We have followed the same procedure and in order to have a closed set of equations, we have integrated with various equations of state, such as pressure vs. energy density. For the present study, we have introduced a reliable estimation for the bulk pressure, for which we have taken into consideration Eckart (first order) and Israel–Stewart (second order) theories for relativistic fluid. For the latter, we found that the resulting differential equations are higher-ordered nonlinear nonhomogeneous so that no analytical solution could be proposed, so far. For the earlier, the only possible solutions relate the Hubble parameter with the scale factor, but none of them could be directly given in terms of the cosmic time.

The present study has a potential to be extended to cover new standard inflationary cosmology with baryosynthesis and dark matter, for which reliable barotropic EoS are unfortunately missing. Taking into consideration the possible influence processes of the beyond standard model is also conditioned to reliable barotropic EoS. Despite that the observational constraints on the cosmological evolution at earlier stages are still challenging, another extension to cover light element abundance with BBN predictions could be a subject for a future study. We would like to suggest concrete predictions and/or observable features of the effects of bulk viscosity in the early cosmological evolution. A framework of new standard cosmology would be rather the ultimate goal. The present script is designed to pave a path towards these goals.

## Figures and Tables

**Figure 1 entropy-23-00295-f001:**
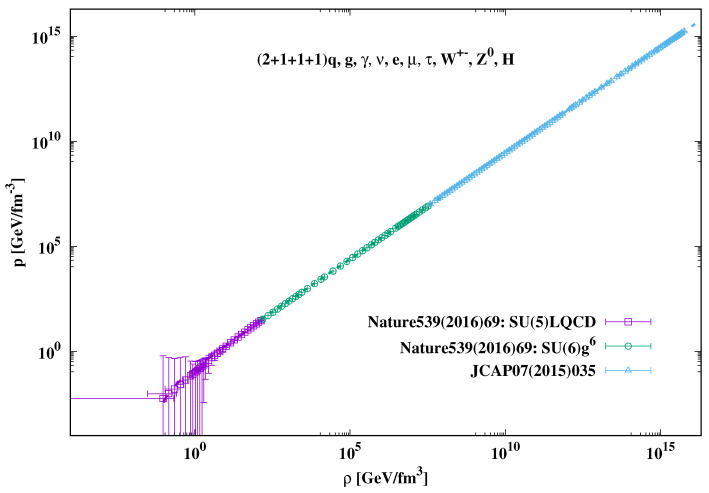
The pressure is depicted as a function of the energy density. Both quantities are given in GeV/fm${}^{3}$ units. The dashed lines present the various parameterizations (see text).

**Figure 2 entropy-23-00295-f002:**
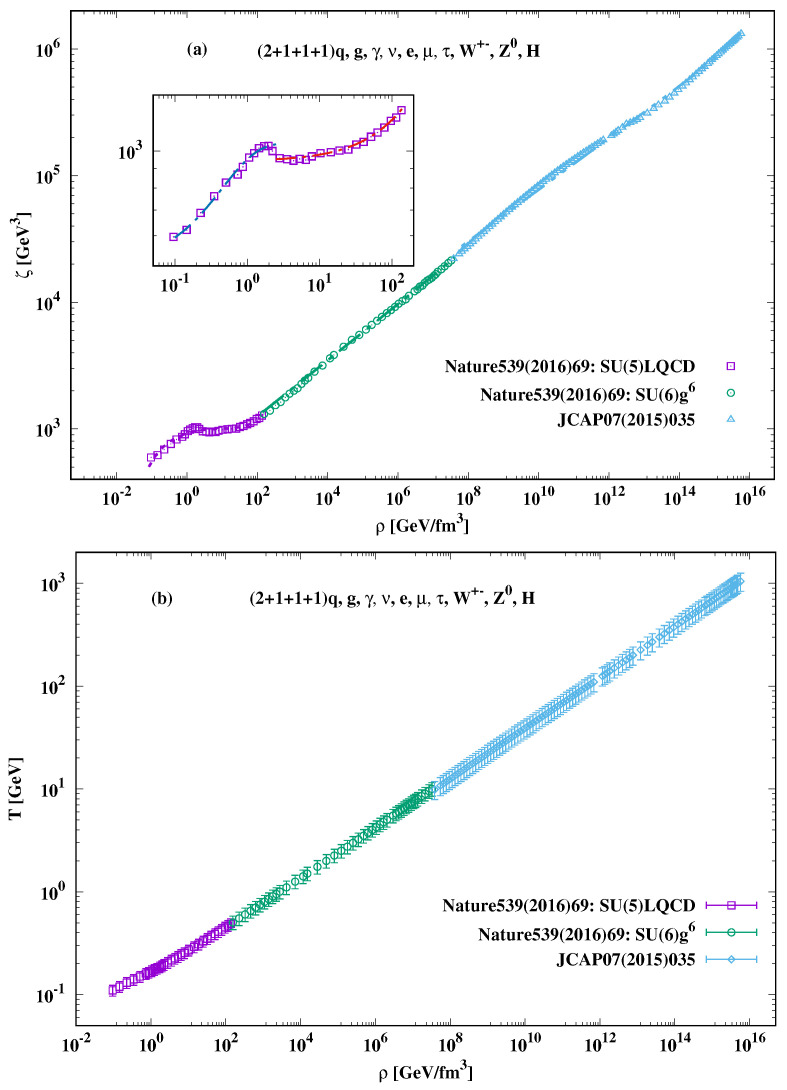
Top panel depicts the energy–density dependence of the bulk viscosity. Bottom panel illustrates the temperature as a function of energy density. The parameterizations are depicted as curves.

**Figure 3 entropy-23-00295-f003:**
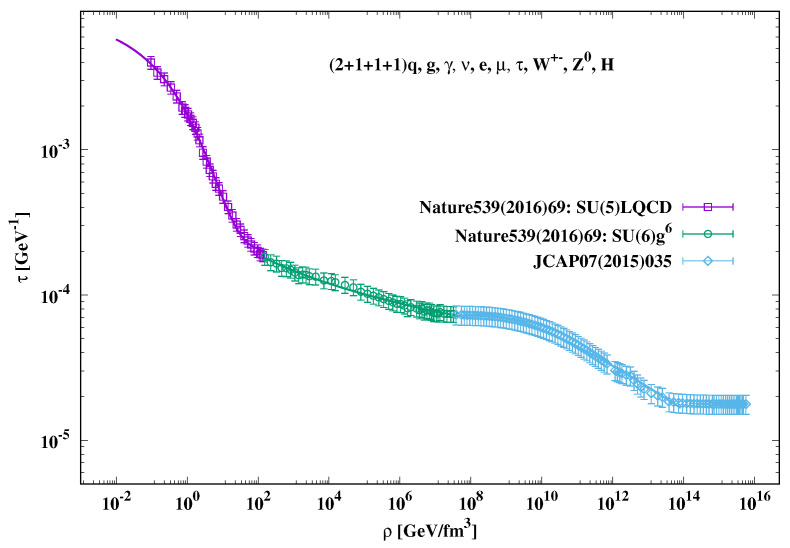
The energy–density dependence of the relaxation time. The parameterizations, Equations (Equation 39)–(41), are depicted as curves.

**Figure 4 entropy-23-00295-f004:**
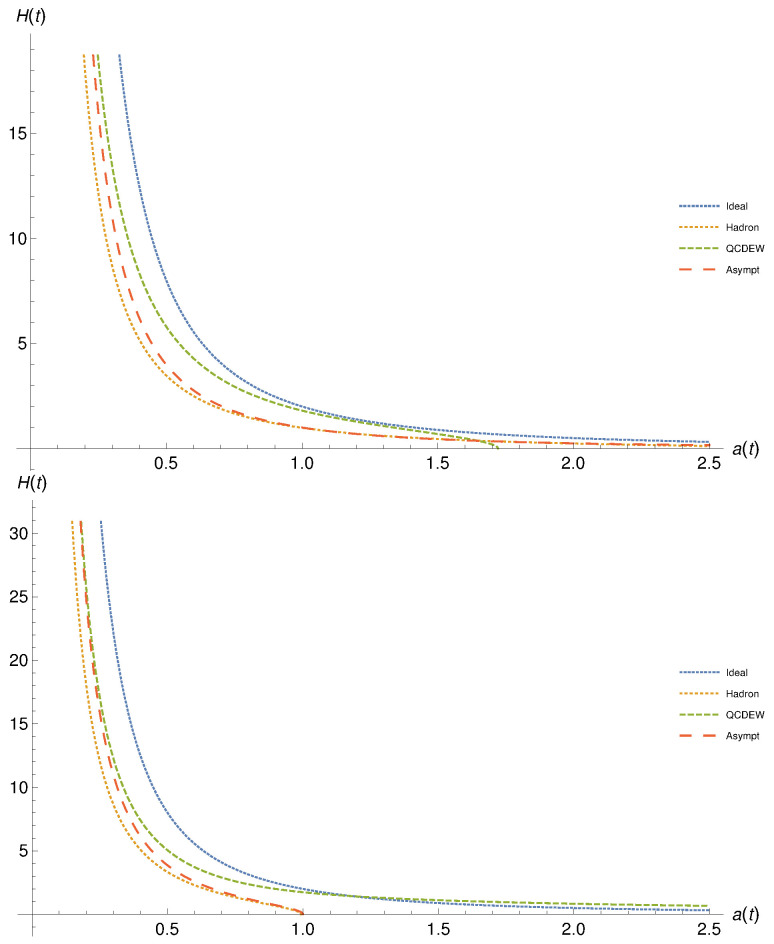
The dependence of the Hubble parameter on the scale factor in non-viscous cosmic background is depicted for finite (top) and vanishing cosmological constant (bottom panel). The equations of state for hadron, QCD-EW, and asymptotic limit are presented as dashed, dotted, and long dashed curves, respectively.

**Figure 5 entropy-23-00295-f005:**
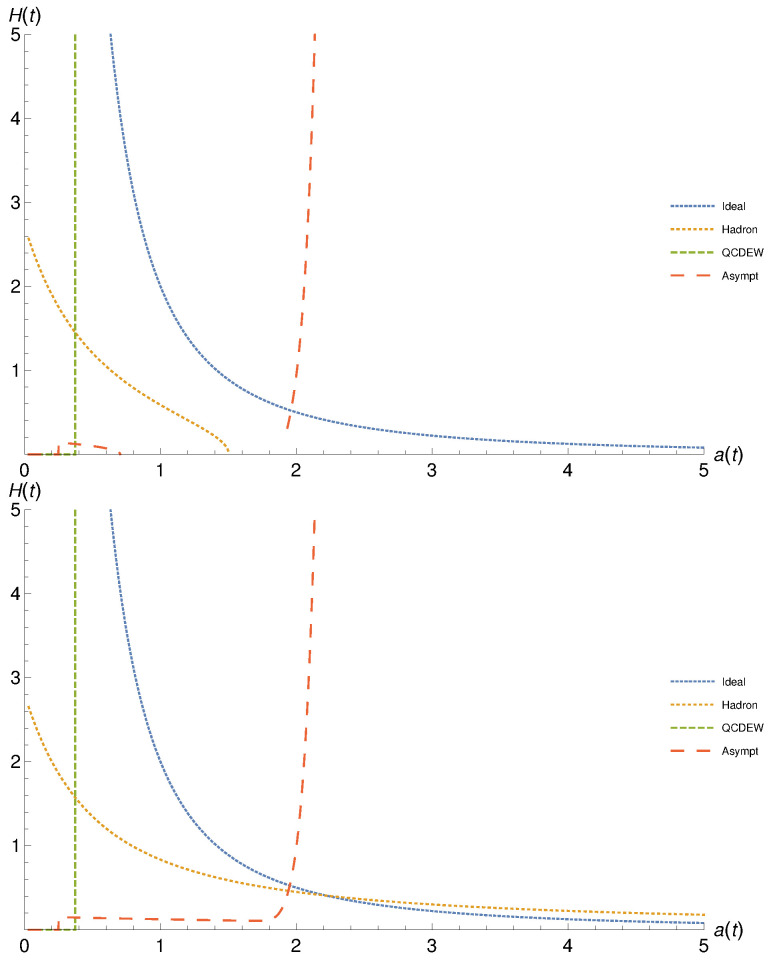
The same as in Figure 4, but here for viscous cosmic geometry (Eckart theory).

**Table 1 entropy-23-00295-t001:** The parameters defining different solutions for Hadron, quantum chromodynamic (QCD)/electroweak (EW) and Asympt. phases corresponding to various equations of state (EoS), Equations (Equation 12), (Equation 21) and (Equation 29), respectively.

	Section	$C_{1}$	$C_{2}$	$C_{3}$	$C_{4}$
Hadron	Section 3.1	$\frac{3}{2}\left(1+\beta_{1}\right)-1$	$4\pi \alpha_{1}-\frac{1}{2}\left(1-\beta_{1}\right)\Lambda$		
QCD/EW	Section 3.2	$\frac{3}{2}\left(1+\beta_{2}\right)-1$	$4\pi \alpha_{2}-\frac{1}{2}\left(1-\beta_{2}\right)\Lambda$	$4\pi {\left(\frac{3}{8\pi k}\right)}^{\delta_{2}}\gamma_{2}$	$\frac{\Lambda }{6}$
Asymp.	Section 3.3	$\frac{3}{2}\left(1+\gamma_{3}\right)-1$	$-\frac{1}{2}\left(1+\gamma_{3}\right)\Lambda$		

## Data Availability

All data are included in the script.

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
