# Peer review of "Early Universe Thermodynamics and Evolution in Nonviscous and Viscous Strong and Electroweak Epochs: Possible Analytical Solutions"

_entropy, 2021, doi:10.3390/e23030295_

Round 1

Reviewer 1 Report

This paper contains interesting ideas and proposals. 

It cannot be published in its present form for the following reasons:

  • many references, e.g. 7,10,11,12,13,30 are incomplete, no journal,
  • no page number, no volume number, only year of publication
  • the text is very poorly written and full of grammatical mistakes, 
  • e.g. in the introduction ;
  • "... could be arose out ...'  "homogeously" all in the first sentences
  • many of the equations are extremely lengthy and probably should be in an appendix

The authors should first present their paper in a manner which is more appropriate for a high level scientific journal.  

Author Response

attachment

Reviewer 2 Report

The paper is strongly based on earlier papers of the authors. The authors discuss the Friedman equation in various epochs of Universe evolution looking for explicit solutons for the Hubble variable. There are some errors and faulty statements which need a correction. Eqs.(12) and (15) are written with parameters C_{1} and C_{2}. If they depend on time then there is no explicit solution. The authors write the solution (19),(20),(32) for a and H in terms of C which follows from (16) if \partial_{t}a=0. So, a =const and H=0. But later on at eqs.(19)(20)(32) the authors say that C depend on time.So all this is inconsistent. Below eq.(22) this is not the Fourier transform.

The paper needs corrections.

Author Response

attachment

Reviewer 3 Report

The paper is dedicated to analysis of possible effect of bulk viscosity on the evolution of very early Universe in the period starting from EW phase transition. Mathematical formalism is developed with the account for first- (Eckart) and second-order (Israel-Stewart) theories for the relativistic cosmic fluid and for integration with viscous equations of state in Friedmann equations. The advantage of this approach is in the use of only known physics of quark-gluon and electroweak processes, studied at the LHC. However, such approach should be properly arranged in the framework of now standard inflationary cosmology with baryosynthesis and dark matter. Physical basis for this standard cosmology is beyond the standard model (BSM) of elementary particles and BSM physics can strongly influence processes in the period considered by the authors and ths possible influence should be taken into account in the the development of their mathematical formalism. Comparison of light element abundance with BBN predictions put stringent constraints on such influence at T< 1 MeV, but observational constraints on the cosmological evolution at earlier stages are still challenging, leaving room for possibly strong effects of BSM physics.

To conclude the proposed formalism would be of interest and practical use for the readers, if it is embedded in the framework of now standard cosmology and can lead to possible observable features of effects of bulk viscosity in the early cosmological evolution.

Author Response

attachment

Round 2

Reviewer 1 Report

The authors have taken into account my previous comments and have considerably improved the appearance of the paper.

It can now be considered in a format fir for  publication.

Reviewer 3 Report

The account for referee's comments by a paragraph in conclusions doesn't correspond to the major revision recommended to authors. The promissed arrangement of their results in the framework of the now standard cosmological scenario should not be postponed for a future work, but should be reflected in the Introduction and the main text in order to specify the meaning of the obtained results in the context of tehmodern cosmology. The paper cannot be accepted in the present form and really major revision of the presentation should be done.
